# ONE STUDENT KNOWS ALL EXPERTS KNOW: FROM SPARSE TO DENSE

## ABSTRACT

Human education system trains one student by multiple experts. Mixture-of-experts (MoE) is a powerful sparse architecture including multiple experts. However, sparse MoE model is easy to overfit, hard to deploy, and not hardware-friendly for practitioners. In this work, inspired by the human education model, we propose a novel task, knowledge integration, to obtain a dense student model (OneS) as knowledgeable as one sparse MoE. We investigate this task by exploring 4 different ways to gather knowledge from MoE to initialize a dense student model, and we then refine the dense student by knowledge distillation. We evaluate our model on both vision and language tasks. Experimental results show, with $3.7\times$ inference speedup, the dense student can still preserve $88.2\%$ benefits from MoE counterpart.

## Introduction

Most people learn from multiple experts in school. Experts from different subjects can help students reach deep understanding and become as knowledgeable as the set of these experts fast (Bransford et al., 1999). Recent study in deep learning proposed mixture-of-experts (MoE), a deep neural network with multiple experts. Each expert is a sub-neural network in the whole model. The key idea of MoE[1] is to divide and conquer the task. MoE encourages each expert to learn from a subset of the input. For each subset of the input, there would be only a sub-network activated. Such sparse computation enables us to scale transformer to trillions of parameters with comparable computation cost (Fedus et al., 2021; Lepikhin et al., 2020; Du et al., 2021).

The MoE-based transformer is powerful and achieved promising results due to its large but sparse-activated model capacity. However, more trainable parameters and sparse conditional computation of MoE introduce overfitting (Xue et al., 2021; Lou et al., 2021). In addition, an extremely large MoE model with trillions of parameters is hard to deploy. Third, MoE is not hardware-friendly, because it needs communication expensive expert parallelism and wasteful sparse tensor representation. Different from MoE, the dense model is widely used but weaker than the sparse model with comparable computation cost. Then, is it possible to combine the strength of sparse and dense model to train a model that is both effective and easy to use?

## Approach

Inspired by human education model, we introduce a new task, *i.e.,* knowledge integration shown in Figure 1. We have two stages in the knowledge integration: (1) knowledge gathering (KG) from MoE; (2) knowledge distillation (KD) to further refine the new dense model (*i.e.,* student). For the first stage, given $E$ experts $\{e_1(\cdot), e_2(\cdot), \ldots, e_E(\cdot)\}$, we are to maximize the knowledge covered in the dense model $s(\cdot)$. Given

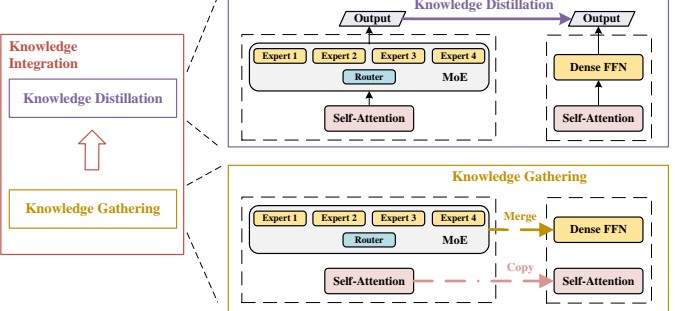

Figure 1: An overview of our general training framework introduced. The overall training framework is knowledge integration, and it includes two stages, gathering and distillation.

[1]More background about MoE can be found in Appendix A.1

input representation $x$, within one transformer block, each expert is an FFN, which can be formulated as $e_i(x) = f_2^i(\sigma(f_1^i(x)))$, where $f_1^i(\cdot)$ and $f_2^i(\cdot)$ and linear transformations of $i^{th}$ expert, $\sigma(\cdot)$ is the activation functions. For the dense student, we have the same architecture as a single expert but different trainable parameters $s(x) = g_2(\sigma(g_1(x)))$. Then, our target is to approximate the trainable parameters of $g_1$ and $g_2$ according to $\{f_1^1, \ldots, f_1^E\}$ and $\{f_2^1, \ldots, f_2^E\}$, respectively. We define this target as gathering from MoE and explore four possible solutions, *i.e.,* summation, averaging, Top-K Knowledge Gathering (Top-KG), and Singular Value Decomposition Knowledge Gathering (SVD-KG). For the Top-KG and SVD-KG, we use Top-K selection or SVD to extract key knowledge from every expert, and then, we merge the key knowledge to initialize the feed-forward network (FFN) layers for a dense student model to approximate the MoE. More details about these four approaches can be found in Appendix A.2. The second stage is fine-tuning the dense student to further minimize the difference between teacher output and student output. We follow the typical KD approaches as our solution.

We define a metric, MoE benefits to measure the ability of a dense student to integrate knowledge from the MoE counterpart, which can be written as $\text{MoE}_{\text{benefits}} = \frac{s_{stu} - s_{dense}}{s_{MoE} - s_{dense}}$, where the score s can be any metric to evaluate the model. For instance, s is accuracy for image classification. The $s_{dense}$ here denotes the dense model's performance without knowledge integration.

### Experiments

We conduct experiments on both language transformer MoE and vision transformer MoE. For vision, we train on ImageNet following Dosovitskiy et al. (2020) and report the top-1 accuracy. For language, we pre-train following BERT (Devlin et al., 2019), and fine-tune on GLUE benchmark Wang et al. (2018) and two versions of SQuAD (Rajpurkar et al., 2016; 2018). We use WideNet (Xue et al., 2021) as the teacher MoE model because its parameters are dominated by MoE layers. As we are the first work, to our best knowledge, focusing on integrating knowledge from a pre-trained MoE, the only two existing strong baselines are the naive knowledge distillation framework proposed in Meta AI MoE Artetxe et al. (2021) and Switch Transformer Fedus et al. (2021). The first one simply initializes the student dense model randomly. The second work initializes the dense model with the non-expert weights. For the weights that cannot be matched (*i.e.,* experts), they train these layers from scratch instead.

|        | Model       | #Para | ImageNet | Benefits(%) |
|--------|-------------|-------|----------|-------------|
| Dense  | ViT-B       | 10M   | 72.8     | -           |
|        | ViT-L       | 15M   | 76.9     | -           |
| Teacher| WideNet-B   | 29M   | 77.5     | -           |
|        | WideNet-L   | 40M   | 79.5     | -           |
| Baseline| Distill-B  | 10M   | 73.8     | 21.3        |
|        | Distill-L   | 15M   | 77.3     | 15.3        |
|        | Switch-B    | 10M   | 74.8     | 42.6        |
|        | Switch-L    | 15M   | 77.8     | 34.6        |
| Ours   | OneS-B Sum  | 10M   | 75.2     | 51.1        |
|        | OneS-L Sum  | 15M   | 78.2     | 48.1        |
|        | OneS-B Avg  | 10M   | 75.3     | 53.2        |
|        | OneS-L Avg  | 15M   | 78.0     | 40.7        |
|        | OneS-B Top-K| 10M   | 75.3     | 53.2        |
|        | OneS-L Top-K| 15M   | **78.4** | **57.7**    |
|        | OneS-B SVD  | 10M   | **75.7** | **61.7**    |
|        | OneS-L SVD  | 15M   | **78.4** | **57.7**    |

Table 1: Top-1 Accuracy and MoE Benefits(%) on ImageNet pre-training. For all models, we share the trainable parameters across transformer blocks for a fair comparison.

|          | Model   | #para | FLOPs | Speedup | SQuAD1.1  | SQuAD2.0  | MNLI | SST-2 | Avg   | Benefits(%) |
|----------|---------|-------|-------|---------|-----------|-----------|------|-------|-------|-------------|
| Dense    | ALBERT  | 12M   | 1.0×  | 3.7×    | 89.3/82.3 | 80.0/77.1 | 81.5 | 90.3  | 84.03 | 0.0         |
| Teacher  | WideNet | 26M   | 2.4×  | 1.0×    | 89.6/82.7 | 80.6/77.4 | 82.6 | 91.1  | 84.71 | -           |
| Baseline | Distill | 12M   | 1.0×  | 3.7×    | 89.4/82.7 | 79.8/76.6 | 81.9 | 90.7  | 84.21 | 26.5        |
|          | Switch  | 12M   | 1.0×  | 3.7×    | 89.5/82.6 | 79.9/77.0 | 82.0 | 90.3  | 84.20 | 25.0        |
| Ours     | OneS SVD| 12M   | 1.0×  | 3.7×    | **89.7/83.0** | **80.2/77.1** | **82.3** | **91.2** | **84.63** | **88.2** |

Table 2: Results of fine-tuning on MNLI, SST-2, and two versions of SQuAD datasets. The FLOPs here means the floating-point operations in FFN layer or MoE layer. We also compare the inference speed on TPU v3-8.

The results can be found in Table 1 and 2. We can observe: (1) SVD-KG performs best in the four KG approaches; (2) OneS can preserve higher MoE benefits than baselines; (3) Compressed models have a significant speedup than MoE counterparts. We also conduct ablation study to verify the effectiveness of different modules, which can be found in Appendix A.5.

URM STATEMENT

The authors acknowledge that at least one key author of this work meets the URM criteria of ICLR 2023 Tiny Papers Track.

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

# A  APPENDIX

## A.1  MIXTURE-OF-EXPERTS

Mixture-of-experts is a typical conditional computation model. In this work, we use a pre-trained MoE model as a teacher, and a dense model as a student to imitate the human education model. Therefore, we briefly review MoE first. Given one MoE model with $E$ trainable experts and input representation $x \in \mathbb{R}^D$, the output of MoE model can be formulated as  Shazeer et al. (2017),:

$$\text{MoE}(x) = \sum_{i=1}^{E} G(x)_i e_i(x) \tag{1}$$

where $e_i(\cdot)$ is a non-linear transformation $\mathbb{R}^D \to \mathbb{R}^D$ of the $i^{\text{th}}$ expert, and $G(\cdot) : \mathbb{R}^D \to \mathbb{R}^E$ is the gating network, $G(x)_i$ is the routing weights of $x$ to the $i\text{-}th$ expert. Usually, both $e(\cdot)$ and $G(\cdot)$ are parameterized by neural networks. Please note the output of $G(\cdot)$ should be activated by softmax function:

$$\text{G}(x) = \text{topK}(\omega(h(x) + \epsilon)) \tag{2}$$

where $\omega$ is the softmax function, $h(\cdot)$ is a linear layer mapping $\mathbb{R}^D \to \mathbb{R}^E$, and $\epsilon \sim \mathcal{N}(0, \frac{1}{E^2})$ is a Gaussian noise for exploration of expert routing. The top-K selection is a key module to activate sub-network sparsely. We usually set $K$ as 1 or 2 for comparable computation cost with the corresponding dense model.

When training MoE model, if we have no regularization, most tokens may be dispatched to a small portion of experts, and other experts receive few tokens. Such an imbalanced assignment would lead to lower efficiency and inferior accuracy Lepikhin et al. (2020); Fedus et al. (2021). Therefore, to achieve balanced workload for different experts, we usually combines router $g(\cdot)$ with load balance loss Lepikhin et al. (2020) $\text{L}_{\text{balance}}$:

$$\text{L}_{\text{balance}} = E \cdot \sum_{i=1}^{E} m_i \cdot P_i \tag{3}$$

where $m$ is a vector and the $i^{\text{th}}$ element of $m$ represents the fraction of tokens dispatched to expert $i$:

$$m_i = \frac{1}{N} \sum_{j=1}^{N} \text{k}(x_j)_i \tag{4}$$

where $N$ is the number of tokens to route, $\text{k}(x_j)$ is an index vector from top-K function. Since the index vector generation here is non-differentiable, we define $P_i$ as:

$$P_i = \omega(h(x) + \epsilon)_i \tag{5}$$

where $P$ is $g(x)$ without top-K routing. When we minimize $\text{L}_{\text{balance}}$, we can see both $m$ and $P$ would close to a uniform distribution.

The trainable router here can also be replaced by non-trainable modules, *e.g.,* BASE layer Lewis et al. (2021). This work focuses on integrating knowledge from a pre-trained MoE instead of MoE variants.

## A.2  DETAILED APPROACH

### A.2.1  KNOWLEDGE GATHERING FROM MOE

We first formulate our KG task. Given an MoE layer with $E$ experts, the target here is to gather knowledge from all experts for one dense student. Each expert comprises two linear layers, and the student shares the same model structure with one single expert. For brevity, we treat each expert as one linear transformation to show our idea, which can be expanded to multiple linear layers easily. For E linear layers $\{f^1, f^2, \ldots, f^E\}$, each linear layer $f^i(\cdot) : \mathbb{R}^{d_1} \to \mathbb{R}^{d_2}$ with weights $W_f^i \in \mathbb{R}^{d_1 \times d_2}$ and bias $b_f^i \in \mathbb{R}^{d_2}$,

$$\begin{aligned} &\text{KG}(f^1, f^2, \ldots, f^E) \\ =&\text{KG}(W_f^1, W_f^2, \ldots, W_f^E; b_f^1, b_f^2, \ldots, b_f^E) \\ \approx&(W_g; b_g) = g \end{aligned} \tag{6}$$

where $g(\cdot) : \mathbb{R}^{d_1} \to \mathbb{R}^{d_2}$ is a linear layer with $W_g \in \mathbb{R}^{d_1 \times d_2}$ and bias $b_g \in \mathbb{R}^{d_2}$.

Before merging the weights, we first initialize $b_g$ from different experts. Since it has much fewer trainable parameters, we simply average the bias vector from different experts:

$$b_g = \frac{1}{E} \sum_{i=1}^{E} b_f^i \tag{7}$$

We employ such a simple policy because knowledge stored in bias is much less than in weights, due to fewer trainable parameters. We justify this assumption by experiments in Appendix A.5.4.

After copying the weights and bias in the perfectly matched layers and averaging bias in the MoE layers, we initialize the dense student model weights by sparse MoE. As the first work focusing on this task, we investigate four methods to gather the knowledge, *i.e.,* summation, averaging, Top-KG and SVD-KG. The first two are the most straightforward methods. We also propose two novel approaches, Top-KG and SVD-KG to extract key knowledge from different experts of a pre-trained MoE.

**Summation and Averaging**   For weights in MoE, we first consider two simple methods. The first one is the summation:

$$W_g = \sum_{i=1}^{E} W_f^i \tag{8}$$

and the second one is averaging:

$$W_g = \frac{1}{E} \sum_{i=1}^{E} W_f^i \tag{9}$$

Although these two gathering methods are simple, as the first work focusing on this task, we investigate them to pave the way for gathering knowledge from MoE models.

**Top-K Knowledge Gathering**   We also propose two novel methods to gather knowledge. For weights, in MoE, a wide over-parameterized model with much more trainable parameters, it is challenging to cover all knowledge in a narrow dense model. Therefore, we have to extract the key knowledge from each expert and then merge them into a single small dense model. Then, the question is, how can we extract the key knowledge of each trainable matrix (*i.e.,* weights)? We first propose Top-K knowledge gathering to extract the sub-matrix of each expert. For $i^{\text{th}}$ expert weight matrix $W^i \in \mathbb{R}^{d_1 \times d_2}$, we calculate the l2 norm of each column as $l^i \in \mathbb{R}^{d_1}$. We then use Top-K selection to pick $K$ columns of $W^i$ according to $l^i$, where $K = \frac{d_2}{E}$. The extracted matrix $W_g^i \in \mathbb{R}^{d_1 \times K}$. Then we concatenate the extracted matrices from all experts as final student initialization $W_g \in \mathbb{R}^{d_1 \times d_2}$.

In practice, since each expert has two linear layers $W^{i_1} \in \mathbb{R}^{d_1 \times d_2}$ and $W^{i_2} \in \mathbb{R}^{d_2 \times d_1}$, there would be a column-mismatch for two extracted matrices from the same expert if we select the sub-matrices of these two matrices independently. To alleviate this issue, we calculate the l2 norm of each column in $W^{i_1}$ and the l2 norm of each row in $W^{i_2}$. The sum of these two l2 norm vectors, *i.e.,* $l^i \in \mathbb{R}^{d_1}$ is fed into Top-K selection and then extract the sub-matrix.

**SVD Knowledge Gathering**   We investigate another novel way to extract key knowledge from experts. Low-rank compression CHen et al. (2021) has shown promising results in capturing key knowledge, which was used to convert a not low-rank matrix to a rank-$k$ decomposition of the weight matrix. Such a low-rank matrix can approximate the knowledge of the whole matrix. On this basis, we can merge the low-rank matrix easier by reconstructing a high-rank matrix from multiple low-rank matrices. Please note, in this work, obtaining rank-$k$ decomposition is not our target. Instead, the rank-$k$ decomposition is just an intermediate step of our decomposing and merging. In this work, we propose to use SVD to extract key knowledge and merge them to initialize another dense matrix:

$$W_f^i = U_f^i S_f^i V_f^{i\,T} \approx U_{f\,K^i}^i S_{f\,K^i}^i V_{f\,K^i}^{i\,T} \tag{10}$$

where $U_f^i \in \mathbb{R}^{d_1 \times d_1}$ and $V_f^i \in \mathbb{R}^{d_2 \times d_2}$ are unitary matrices, $S_f^i \in \mathbb{R}^{d_1 \times d_2}$ is a diagonal matrix. We usually select the top-K elements in $S_f^i$ and then construct $U_{f\,K^i}^i \in \mathbb{R}^{d_1 \times K^i}$, $S_{f\,K^i}^i \in \mathbb{R}^{K^i \times K^i}$ and $V_{f\,K^i}^i \in \mathbb{R}^{d_2 \times K^i}$ to approximate $W_f^i$.

When $k$ is fixed, every matrix has the rank-$k$ decomposition to approximate the original matrix. However, we cannot guarantee the key knowledge in every expert can be covered by a fixed rank-$k$ decomposition. Thus, we define an adaptive SVD ratio $\lambda \in (0,1]$ to ensure:

$$\rho(S_{f\,K^i}^i) \approx \lambda \rho(S_f^i) \tag{11}$$

where $\rho(S_f^i)$ denotes the sum of diagonal elements of $S_f^i$. If $\lambda = 1$, all ranks would be preserved for a full-rank matrix. We then collect the decomposition of each expert and concatenate them as:

$$
\begin{aligned}
[\,U_g\,] &= \begin{bmatrix} U_{f\,K^1}^1 & \cdots & U_{f\,K^E}^E \end{bmatrix}, \\
[\,S_g\,] &= \begin{bmatrix} S_{f\,K^1}^1 & & \\ & \ddots & \\ & & S_{f\,K^E}^E \end{bmatrix}, \\
[\,V_g\,] &= \begin{bmatrix} V_{f\,K^1}^1 \\ \vdots \\ V_{f\,K^E}^E \end{bmatrix}
\end{aligned}
\tag{12}
$$

We can then obtain $W_g$ as:

$$W_g = U_g S_g V_g{}^T \tag{13}$$

$W_g$ is a rank-$K_g$ matrix, where $K_g = \Sigma_{i=1}^E K^i$, covering the key knowledge of every expert.

After SVD-KG, knowledge has been integrated from pre-trained MoE. However, during knowledge gathering, it is unavoidable to induce noise when we remove conditional computation. Detailed analysis of the induced noise during gathering can be found in Appendx A.3.

### A.2.2 KNOWLEDGE DISTILLATION

To mine the knowledge from noise, we adopt soft knowledge distillation Hinton et al. (2015) to fine-tune the dense student. Soft distillation minimizes the Kullback-Leibler divergence between the output of the teacher and the student. The corresponding distillation loss can be written as:

$$\mathrm{L}_{distill}^{soft} = T^2 \mathrm{L}_{\mathrm{KL}}(\omega(z_s/T), \omega(z_t/T)) \tag{14}$$

where $\omega$ is the softmax function, $L_{KL}$ is Kullback-Leibler divergence loss, $z_s$ and $z_t$ are the logits of student and teacher, respectively, and $T$ is the softmax temperature.

### A.2.3 OPTIMIZATION

Our final loss function is simple:

$$\mathrm{L}_{\mathrm{total}} = \alpha \mathrm{L}_{\mathrm{main}} + (1 - \alpha)\mathrm{L}_{\mathrm{distill}} \tag{15}$$

where $\alpha$ is used to balance the main loss and the distillation loss. The main loss depends on the task. For instance, to classify images, it is cross-entropy. For BERT pre-training, it should be the masked language modeling loss and next sentence prediction loss. The distillation loss here can be either soft distillation loss or hard-label distillation loss. Since our pre-trained MoE is fixed during knowledge distillation, we do not need the load balance loss of MoE-based transformer.

### A.3 KNOWLEDGE GATHERING NOISE ANALYSIS

We are to discuss and analyze the induced noise during SVD knowledge gathering in this section.

Given one MoE layer $\mathrm{MoE}(\cdot)$, the target of SVD-KG is to integrate its knowledge to a dense layer $g(\cdot)$ in the student model. For brevity, we set every expert and the dense student layer as the single linear layer. There are $E$ experts in MoE layer: $\{f^1, \ldots, f^E\}$ with weights $\{W_f^1, \ldots, W_f^E\}$ and

bias $\{b_f^1, \ldots, b_f^E\}$. The dense student layer is $g$ with weights $W_g$ and bias $b_g$. According to Eq. 1, the MoE layer can be written as:

$$\begin{aligned} \text{MoE}(x) &= \sum_{i=1}^{E} G(x)_i e_i(x) \\ &= \sum_{i=1}^{E} p_i h_i (W_f^i x + b_f^i) \end{aligned} \tag{16}$$

where $p$ is the routing score of router, $h$ is an index vector. For the selected experts, $h_i = 1$, and $h_i = 0$ for other unselected experts. Due to the load balance loss during MoE training, we can assume $p_i \approx 1.0$ when $h_i = 1$. Then, we can approximate MoE layer by SVD:

$$\begin{aligned} \text{MoE}(x) &= \sum_{i=1}^{E} p_i h_i (U_f^i S_f^i V_f^{i^T} x + b_f^i) \\ &\approx \sum_{i=1}^{E} h_i (U_{f\,K^i}^i S_{f\,K^i}^i V_{f\,K^i}^{i^T} x + b_f^i) \\ &\approx \sum_{i=1}^{E} h_i \sum_{j=1}^{K^i} u_{f\,K^i}^{ij} s_{f\,K^i}^{ij} v_{f\,K^i}^{ij^T} x + \sum_{i=1}^{E} h_i b_f^i \end{aligned} \tag{17}$$

where $K^i$ is the selected rank of $i$-th expert.

According to Eq. 13, $g(\cdot)$ can be formulated as:

$$g(x) = \sum_{i=1}^{E} \sum_{j=1}^{K^i} u_{f\,K^i}^{ij} s_{f\,K^i}^{ij} v_{f\,K^i}^{ij^T} x + \frac{1}{E} \sum_{i=1}^{E} b_f^i \tag{18}$$

For brevity, to analyze, we assume MoE layer here is to select the 1-st expert, and then the MoE layer can be written as:

$$\text{MoE}(x) \approx \sum_{j=1}^{K^1} u_{f\,K^1}^{1j} s_{f\,K^1}^{1j} v_{f\,K^1}^{1j^T} x + b_f^1 \tag{19}$$

and the student dense layer:

$$\begin{aligned} g(x) &= \sum_{j=1}^{K^1} u_{f\,K^1}^{1j} s_{f\,K^1}^{1j} v_{f\,K^1}^{1j^T} x + b_f^1 \\ &+ \sum_{i=2}^{E} \sum_{j=1}^{K^i} u_{f\,K^1}^{ij} s_{f\,K^1}^{ij} v_{f\,K^1}^{ij^T} x \\ &+ \frac{1}{E} \sum_{i=2}^{E} b_f^i - \frac{E-1}{E} b_f^1 \end{aligned} \tag{20}$$

Since the non-selected experts do not interact with the current input token $x$, we assume, for the non-selected experts, we let $\epsilon_1 = f^i(x)$ and $\epsilon_1 \sim \mathcal{N}(\mu_1, \sigma_1^2)$ and $\epsilon_2 = b_f^i x$ and $\epsilon_2 \sim \mathcal{N}(\mu_2, \sigma_2^2)$. According to Eq. 11, $g(x)$ can be written as:

$$g(x) = \sum_{j=1}^{K^1} u_{f\,K^1}^{1j} s_{f\,K^1}^{1j} v_{f\,K^1}^{1j^T} x + \lambda [(E-1)\epsilon_1 - \frac{E-1}{E} \epsilon_2] \tag{21}$$

The low-rank approximation ensures $\sum_{j=1}^{K^1} u_{f\ K^1}^{1j} s_{f\ K^1}^{1j} v_{f\ K^1}^{1j\ T} + b_f^1$ cover most informative knowledge in the selected expert, and noise reduced linearly along $\lambda$. When we are integrating knowledge from experts, a smaller $\lambda$ is required to reduce noise.

## A.4 HYPER-PARAMETERS

### A.4.1 COMPUTER VISION

Table 3: Hyper-parameters on ImageNet pre-training and Cifar10 finetuning. $\alpha$ and $\lambda$ are from Eq. 15 and Eq. 11

| Parameter | ImageNet | Cifar10 |
| --- | --- | --- |
| Epoch | 300 | 100 |
| Warmup Epochs | 30 | 0 |
| Batch Size | 4096 | 512 |
| Learning rate | 0.004 | 0.03 |
| Weight Decay | 0.1 | 0 |
| Dropout | 0.1 | 0.1 |
| Label smoothing | 0.1 | 0 |
| Mixup prob. | 0.5 | 0.5 |
| $\alpha$ | 0.25 | - |
| $\lambda$ | 0.75 | - |

Most hyper-parameters are set following existing works (*e.g.,* ViT, WideNet). The main difference is the learning rate. Since we are training from a dense model initialized by a MoE model. We observe that a large learning rate harms accuracy. We, therefore, set a smaller learning rate as 0.004 (0.01 in WideNet).

### A.4.2 NATURAL LANGUAGE PROCESSING

Table 4: Hyper-parameters on NLP downstream tasks fine-tuning.

| Parameter | SQuAD1.1/2.0 | MNLI | SST2 |
| --- | --- | --- | --- |
| Steps | 3649/8144 | 10000 | 5234 |
| Warmup | 365/814 | 1000 | 314 |
| Batch Size | 48 | 128 | 128 |
| LR | 5e-5/3e-5 | 3e-5 | 4e-5 |
| Dropout | 0/0 | 0 | 0 |
| Max Length | 384/512 | 512 | 512 |

We follow the hyper-parameters in Devlin et al. (2019); Lan et al. (2019); Xue et al. (2021) and the final hyper-parameters are reported in Table 4.

## A.5 ABLATION STUDY

### A.5.1 ABLATION ON THE CONTRIBUTIONS OF KG AND KD

The first ablation study is to investigate the contributions of knowledge gathering and knowledge distillation. As shown in Table 5, there is a significant performance drop without knowledge gathering, which shows the knowledge included in pre-trained sparse model is critical to improve the student model's performance. For the model without KD, in this experiment, we adopt the $L_{main}$ in Eq. 15 as the only loss function. We can see the knowledge distillation is helpful, as the prediction of teacher can instruct the student to mine knowledge in noisy weights gathered. In addition, when the dense model does not gather knowledge from MoE, the KD enables the training process of the

| Model | ImageNet |
|---|---|
| OneS-B | 75.7 |
| w/o KG | 73.8 |
| w/o KD | 75.0 |
| w/o KG & KD | 72.8 |
| OneS-L | 78.4 |
| w/o KG | 77.3 |
| w/o KD | 77.6 |
| w/o KG & KD | 76.9 |

Table 5: Top-1 Accuracy of ablation study on ImageNet to investigate the contributions of knowledge gathering (KG) and knowledge distillation (KD). The KG here is using SVD-KG, and the KD here is using soft-distillation, as we found they perform better by investigation.

lite model (*i.e.,* OneS-B) more stable. For the large model, removing both knowledge gathering and knowledge distillation will also harm the performance.

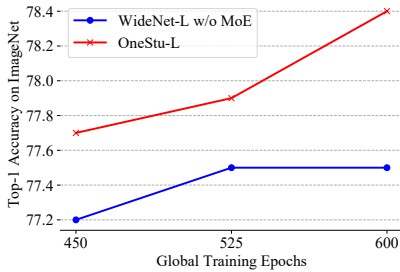

Figure 2: Top-1 Accuracy of ablation on ImageNet to investigate the contribution of more global training epochs.

### A.5.2 ABLATION ON TOTAL TRAINING STEPS

Since we conduct two stages of training in our framework, the total training steps of OneS are more than the dense model trained from scratch without distillation. The second set of ablation study is to verify whether the improvement of our model is from more training iterations. To this end, we train the OneS without KG and KD from scratch for comparable global training epochs. We use OneS-L as a platform for this set of experiments because we observe the unstable training of OneS-B without both KG and KD. As shown in Figure 2, when training with comparable global epochs, our OneS outperforms baselines by a large margin consistently. Also, when scaling to more epochs, WideNet without MoE stops to improve, but our OneS can still obtain benefits from more training. We also investigate two types of knowledge distillation approaches, soft distillation Hinton et al. (2015) and hard-label distillation Touvron et al. (2021). The last set is to ablate the SVD ratio $\lambda$.

### A.5.3 ABLATION STUDY ON SVD RATIO

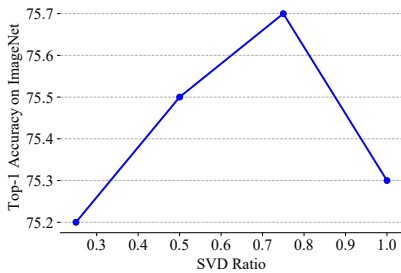

Figure 3: Top-1 Accuracy of ablation on ImageNet to ablate the SVD ratio $\lambda$.

We also conduct ablation study on SVD ratio $\lambda$, which denotes the ratio of selected k. As shown in Figure 3, when $\lambda = 0.75$, OneS-B achieves sweet point.

### A.5.4 EXPERIMENTAL JUSTIFICATION FOR LESS KNOWLEDGE IN BIAS

Table 6: Top-1 Accuracy of MoE model without bias.

| Approach | ImageNet |
|---|---|
| WideNet-B | 77.5 |
| WideNet-B w/o bias | 77.3 |

We re-trained the teacher MoE model (*i.e.,* WideNet-B) without bias in MoE layer. As shown in Table 6, we found that there is no obvious performance drop. That is, the bias in MoE layer has little impact on results, which means there is less knowledge than weights.

## A.6 RELATED WORK

### A.6.1 MIXTURE-OF-EXPERTS

MoE has shown promising results on various tasks. Recent works scaled a dense model to a sparse one by MoE. Faster convergence speed of MoE can save the global computation cost. One typical way to use MoE is, by replacing the FFN layer in transformer Vaswani et al. (2017) by an MoE layer. Lepikhin *et al.* Lepikhin et al. (2020) first scale machine translation transformer model to 600 million parameters using automatic sharding. After that, Fedus *et al.* Fedus et al. (2021) further scales the transformer to trillion parameter models with simple and efficient sparsity and shows promising results on natural language understanding. In computer vision, ViT-MoE Ruiz et al. (2021) matches SoTA performance on ImageNet using 14.7 billion of parameters, while requiring as little as half of the computation at inference time. Recent work Lou et al. (2021) investigated the MoE usage on MLP-Mixer, which also achieved better effectiveness and efficiency than the dense model. Instead of scaling up, this work uses and fixes the pre-trained MoE model. The core target is to combine the effectiveness of MoE and the usability of dense model.

### A.6.2 KNOWLEDGE INTEGRATION

Knowledge inheritance Qin et al. (2021) is related to our knowledge integration. Knowledge inheritance usually inherits knowledge from small pre-trained model and then speed-up the training of large models. Contrastively, our work is integrating knowledge from a large MoE model. Sun *et al.* Sun et al. (2019) proposed to integrate knowledge by using knowledge masking strategies. Please note our knowledge integration is different from theirs. Instead of a self-supervised learning approach to integrate knowledge from data, our work is to integrate knowledge from pre-trained MoE. There are also a few works focusing on inheriting knowledge from a dense model to initialize a MoE model, which can be seen as an inverse process of ours. For instance, Zhang *et al.* Zhang et al. (2022) duplicated dense model multiple times to initialize MoE models. Zhang *et al.* Zhang et al. (2021) proposed MoEfication. The proposed approach is to inherit knowledge from a dense model and obtain an MoE model with comparable parameters to reduce the computation cost. In general, MoEfication is a sparsification approach. In Switch Transformer Fedus et al. (2021), authors tried to initialize trainable parameters except for MoE layers to speed-up MoE training, although their main purpose is to scale transformer to trillions of parameters.

### A.6.3 KNOWLEDGE DISTILLATION

Knowledge distillation is a well-studied problem. One related direction is multi-teacher knowledge distillation (You et al., 2017). For instance, Chebotar & Waters (2016) distill from multiple teachers for speech recognition. Yuan et al. (2021) develop a reinforced method to dynamically assign weights to teacher models for different training instances and optimize the performance of student model. However, our task is different from the multi-teacher knowledge distillation in some important aspects. First, the knowledge integration has only one teacher. This teacher is a single model

with many experts in MoE layers. In MoE-based transformer, each expert is an FFN layer. As a comparison, the multi-teacher distillation has many teacher models and each of them is a complete model including many layers. Second, the MoE model has similar architecture in many layers but also sparse architecture in MoE layers. For instance, the trainable parameters in attention layer can be perfectly mapped into the student model. This provides a great potential to not only distill through model outputs but also map the model weights directly. Lastly, in multi-teacher KD, the teacher model typically has a higher computation cost compared to the student model, while the computation cost is usually similar in MoE-based KD. How to leverage the matching computation cost and large parameter gap is also another valuable research question.

