# OpenReview forum: "One Student Knows All Experts Know: From Sparse to Dense"
_ICLR.cc/2023/TinyPapers — Submitted to Tiny Papers @ ICLR 2023_

### Official Review · Reviewer_rQaJ · 2023-03-31

**Confidence:** 4

**Summary Of Contributions:**

The authors propose a novel task of knowledge integration to obtain a dense student model which is comparable to a sparse MoE.

**Rating:**

Clear, Correct, and Reproducible (CCR): a submission which meets the reviewing criteria

**Strengths And Weaknesses:**

Strengths:
- The problem is relevant, well formulated and clearly explained.
- The experiments are quite detailed covering both vision and language MoE.
- Data/hyperparameters are mentioned for reproducibility.

Weakness:
- The formatting and structure of the paper needs to be fixed.
- In a related point, although the paper is well written, it is a bit too dense for this track which has a 2 page limit for main text.

**Suggested Changes:**

Please refer to point 1 of the weakness above.

---

### Official Review · Reviewer_gcxZ · 2023-04-01

**Confidence:** 3

**Summary Of Contributions:**

This paper tackles the problem of distilling a faster/smaller model from multiple expert models. It demonstrations good results in a inference speedup in addition to retaining most of the original expert performances.

**Rating:**

Clear, Correct, and Reproducible (CCR): a submission which meets the reviewing criteria

**Strengths And Weaknesses:**

Strengths:
- There is a clear identification of current issues suffered by MoE-based transformer models, and this paper shows good results that tackle these issues (lower inference time while retaining performance)
- Apart from minor grammar issues paper is well written with good figures and tables that help highlight the results and explain the approach well.
- Appendix appears to be well written and documented, so reproducibility should be possible (open-sourced code would be great though)

Weaknesses:
- The benefits metric could be explained a bit more clearly. It's not immediately clear what $s_{stu}$ might refer to.
- the input representation $x$ could be clarified. Is this the input from the datasets tested on? The word representation makes it sound that the original dataset inputs were processed/formatted into some other representation.

**Suggested Changes:**

Minor:
Table 2 description can be improved by explaining what the bolds mean (usually it means best result but because of the existence of the teacher this is not exactly true).
Same for table 1, although this one is more confusing since there are multiple bolds in one column across a few models.

Grammatical (not super important! but always helpful):
"Recent study in deep learning proposed mixture-of-experts (MoE), a deep neural
network with multiple experts." -> "A recent study in deep learning proposed mixture-of-experts (MoE), a deep neural
network with multiple experts." Also perhaps this is a typo? MoE is not a deep neural net, but a ML technique.

"Inspired by human education
model" -> "Inspired by the human education
model..."

---

### Meta-Review · Area_Chair_zbZG · 2023-04-07

**Recommendation:** Invite to present
**Confidence:** 4

**Metareview:**

1. Clarity: The paper provides appropriate literature and communication is clear
2. Correctness: The findings are justified aptly
3. Reproducibility: Through the mathematically approach is explained well in appendix, reproducible code & data is  not provided
4. Pros:
    * 4 gathering techniques used for stage 1 of knowledge integration(summation, averaging, Top-K knowledge gathering, SVD Knowledge gathering
    * Experiments conducted for both language & vision transformer MoE
    * Optimization: Hyperparameters for both NLP & CV is discussed
    * Ablation study on knowledge gathering and knowledge distillation
5. Cons:
    * This paper is 2 page Short version of original paper published in Jan2022, this paper fails to provide credit in references to original paper [One Student Knows All Experts Know: From Sparse to Dense](https://arxiv.org/pdf/2201.10890.pdf)
    * Reproducible code & data not provided




**Summary:**

Introduced novel task, knowledge integration with 2 stages: knowledge gathering from MoE & Knowledge distillation to refine new dense model

**Comments And Feedback To The Authors:**

Minor revision of including the original paper in the references
1. This paper is a short 2 page version of paper [One Student Knows All Experts Know: From Sparse to Dense](https://arxiv.org/pdf/2201.10890.pdf) Cite the original paper in the references
https://arxiv.org/pdf/2201.10890.pdf ( or ) Published semanticscholar: https://www.semanticscholar.org/paper/One-Student-Knows-All-Experts-Know%3A-From-Sparse-to-Xue-He/9caa83288602f6c3734c78d8d89bb358b263da24
2. Reproducible Code & data should be added in appendix, this is one the major  review criteria for empirical findings
3. URLs placed in footer can be moved to reference session


**Reason For Not Giving A Higher Recommendation:**

Following revisions are required in this paper
1. This paper is 2 page Short version of original paper published in Jan2022, this paper fails to provide credit in references to original paper [One Student Knows All Experts Know: From Sparse to Dense](https://arxiv.org/pdf/2201.10890.pdf)
   * Published semanticscholar: https://www.semanticscholar.org/paper/One-Student-Knows-All-Experts-Know%3A-From-Sparse-to-Xue-He/9caa83288602f6c3734c78d8d89bb358b263da24
2. Reproducible code & data not provided




**Reason For Not Giving A Lower Recommendation:**

N/A

---

### Decision · Program_Chairs · 2023-04-08

**Decision:**

No revision received; not invited to archive

**Comment:**

It's been pointed out to the PCs that this is a shortened version of a published work (https://arxiv.org/abs/2201.10890). Not enough evidence was given to support that this version is sufficiently different from the original (despite the difference in length), and the submission didn't make the fact clear (didn't cite or mention the other published work). We have hence come to a conclusion to give a "invite to revise" rating.

---

> ### Author Response · Authors · 2023-04-13
> **This work has not been published**
>
> I would like to express my sincere gratitude for organizing the ICLR Tiny Paper and for taking the time to review our submission.
>
> We would like to clarify that while the preprint is available on arXiv, it has not been published or accepted for publication by any other conference or journal. We understand that the ICLR review process requires that submissions be original work that has not been previously published, and we assure you that our paper meets this requirement.
>
> We acknowledge that the preprint was not explicitly mentioned in our submission. Actually, we did not mention the preprint because we want to ensure a double-blind review process. We apologize for any confusion this may have caused, and we would be grateful for the opportunity to present it at ICLR. We would be happy to address any further questions or concerns you may have.
>
> Thank you once again for your time and consideration.
>
> Best,
> Authors